# Substance Abuse and Male Hypogonadism

**DOI:** 10.3390/jcm8050732

**Published:** 2019-05-22

**Authors:** Ylenia Duca, Antonio Aversa, Rosita Angela Condorelli, Aldo Eugenio Calogero, Sandro La Vignera

**Affiliations:** 1Department of Clinical and Experimental Medicine, University of Catania, 95123 Catania, Italy; ylenia.duca@gmail.com (Y.D.); rosita.condorelli@unict.it (R.A.C.); acaloger@unict.it (A.E.C.); 2Department of Experimental and Clinical Medicine, University Magna Graecia of Catanzaro, 88100 Catanzaro, Italy; aversa@unicz.it

**Keywords:** hypogonadism, oligozoospermia, substance abuse, drug abuse, alcohol, cigarette smoking, cannabis, amphetamines, opioids, anabolic-androgenic steroids

## Abstract

Progressive deterioration of male reproductive function is occurring in Western countries. Environmental factors and unhealthy lifestyles have been implicated in the decline of testosterone levels and sperm production observed in the last fifty years. Among unhealthy lifestyles, substance and drug abuse is a recognized cause of possible alterations of steroidogenesis and spermatogenesis. Alcohol, opioids and anabolic-androgenic steroids are capable to reduce testosterone production in male interfering with testicular and/or hypothalamic-pituitary function. Other substances such as nicotine, cannabis, and amphetamines alter spermatogenesis inducing oxidative stress and subsequent apoptosis in testicular tissue. Substance and drug abuse is a potentially reversible cause of hypogonadism, defined as the failure of the testis to produce physiological concentrations of testosterone and/or a normal number of spermatozoa. The identification of the abuse is important because the withdrawal of substance intake can reverse the clinical syndrome. This review summarizes the most important clinical and experimental evidence on the effect of substance abuse on testosterone and sperm production.

## 1. Introduction

According to the most recent definition, hypogonadism is a clinical syndrome resulting from the failure of the testis to produce physiological concentrations of testosterone and/or a normal number of spermatozoa due to pathologies of the hypothalamic–pituitary–testicular axis [1].

Pathogenic noxae acting at testicular level give rise to a syndrome characterized by low testosterone concentrations, impairment of spermatogenesis, and elevated gonadotropin levels, defined as primary hypogonadism. Otherwise, secondary hypogonadism is caused by a dysfunction of the hypothalamus–pituitary unit, resulting in low testosterone concentrations, impairment of spermatogenesis, and low or inappropriately normal gonadotropin levels [1]. Hypogonadism is defined as organic when a congenital, structural, or destructive disorder results in a permanent dysfunction, and functional when potentially reversible conditions suppress temporarily gonadotropin and/or testosterone production [1].

The diagnosis of hypogonadism can be formulated when at least two testosterone values below normal are found in patients with symptoms and signs of testosterone deficiency. Testosterone must be measured in morning samples after a fast of at least 8 hours. To uniform reference values among laboratories and assays, the Endocrine Society recently proposed a harmonized reference range, according to which total testosterone values should be considered reduced if lower than 264 ng/dL or 9.2 nmol/L [1]. The measurement of free testosterone is not routinely recommended; however, it can be useful in patients with altered sex hormone-binding globulin (SHBG) concentrations. Indeed, only 2–4% of testosterone circulates in free form, while the remainder is bound to SHBG and, less tenaciously, to albumin. In conditions when SHGB levels are reduced (e.g., obesity, type 2 diabetes mellitus, androgen use), total testosterone concentrations could be below normal values even if free testosterone levels are in the normal range. The gold standard for free testosterone measurement is the equilibrium dialysis method. However, most laboratories use immunoassays that are less accurate. In this case, to calculate free testosterone starting from total testosterone, SHBG, and albumin values is preferable [1].

Symptoms and signs suggestive of testosterone deficiency include reduced libido and sexual activity, decreased spontaneous erections, erectile dysfunction, gynecomastia, infertility, osteopenia/osteoporosis, hot flushes, and sweats. Other non-specific symptoms, such as fatigue, depressed mood, poor concentration and memory, sleep disturbance, reduced muscle mass and strength, increased body fat and body mass index [1], although not decisive for defining the diagnosis, are important from a clinical point of view because they have a great impact on the quality of life of hypogonadal patients.

The prevalence of hypogonadism is rising over time, not only for the improvement in diagnostic procedures. In 2007, Travison and colleagues analyzed data from the Massachusetts Male Aging Study (MMAS) about testosterone concentration and observed an overtime decline in total testosterone levels higher than that attributable to aging. The Author hypothesized that the recorded age-independent population decrease in testosterone could be attributable to birth cohort differences or to environmental factors [2]. The same trend has been shown in Europe. Data from Danish population surveys conducted from 1982 to 2001 evidenced a secular trend in testosterone and SHBG serum levels among age-matched men, with lower levels in the younger men and in those more recently studied [3].

The progressive deterioration of male reproductive function also affects sperm production. In 2017, Levine and colleagues performed a systematic review and meta-regression analysis, showing that sperm concentration in Western countries declined by 1.4% per year, with an overall decline of 52.4%, between 1973 and 2011. Similarly, total sperm count in the same period declined by 1.6% per year, with an overall decline of 59.3%. No trend in concentration and total sperm count reduction was found in other countries [4].

Factors that could contribute to the worsening of testicular function in Western countries include the progressive increase in visceral adiposity among the population, changes in lifestyle and behaviors, environmental pollution and exposure to endocrine-disrupting compounds (i.e., phthalates) [5]. Among unhealthy behaviors, alcohol abuse, cigarette smoking, excessive caffeine intake, illicit drug intake, opioid consumption, and inappropriate use of anabolic steroids have been studied as a possible cause of reduced sperm production and/or reduced testosterone levels in hypogonadal men [6].

In our review, we summarized the most important clinical and experimental evidence on the effect of substance abuse on testosterone and sperm production For many of the substances examined most of the evidence comes from in vivo and in vitro animal studies or from retrospective human studies. Indeed, for ethical reasons, no intervention studies on humans can be performed. Unfortunately, data obtained in animals are not always reproducible in humans; so some aspects regarding the mechanisms of action of several substances on the reproductive function must be further clarified.

## 2. Alcohol

Since ancient times the consumption of alcoholic beverages has been part of the socio-cultural heritage of most populations. However, chronic and acute alcohol abuse is involved in the pathogenesis of many diseases, including liver diseases, cancers, cardiovascular disease, and neuropsychiatric disorders. The effects of alcohol intake on male reproductive function have also been evaluated, in vitro and in vivo. Both testosterone production and spermatogenesis seem to be affected by alcohol abuse in a dose-dependent manner: heavy drinkers are more likely to have a poor testicular function than moderate consumers.

### 2.1. Effects on Testosterone Production

It has been demonstrated that alcohol could decrease testosterone blood concentration acting both on testicular and central (hypothalamic and pituitary) level [7].

Since the 1980s, it is known that ethanol and acetaldehyde are important Leydig cells toxins. Van Thiel and colleagues performed both in vivo and in vitro studies assessing the alcohol effect on Leydig cells function [8]. In their in vivo human study, they evaluated the hormonal status of chronic alcoholic men, comparing it with that of healthy volunteers to whom a quantity of ethanol corresponding to a pint of whiskey/day was given for 30 days. They showed that almost all alcoholic men had low-normal or low testosterone levels and increased gonadotropin levels. In the healthy male volunteers, testosterone levels began to reduce from the baseline after 72 hours of ethanol ingestion and reached levels similar than those of alcoholic men after 30 days, while gonadotropins remained in the normal range [8]. In their in vivo animal studies, they showed that alcohol-fed rats had testosterone levels reduced by half compared to isocaloric non-alcohol fed rats [8]. In in vitro studies, they showed that rat testes perfused with ethanol and acetaldehyde showed a reduced production and secretion of testosterone in a dose-dependent manner. The same phenomenon occurred in cultured rat Leydig cells [8].

The reduced production of testosterone by Leydig cells is due to the inhibitory action of alcohol on the enzymes 3β-hydroxysteroid dehydrogenase and 17-ketosteroid reductase, which catalyze respectively the conversion from pregnenolone to progesterone and from androstenedione to testosterone. Progesterone is the precursor for the synthesis of testosterone, so its lack could lead to a decreased production of testosterone [9]. Furthermore, alcohol enhances the production of radical oxygen species that suppress the expression of the steroidogenic acute regulatory protein (StAR), which regulates the rate-limiting step in the steroid hormone biosynthesis that is the transport of cholesterol from the outer to inner mitochondrial membrane [10].

The decreased testosterone concentration in heavy drinkers depends on the reduced production by Leydig cells but also on the increased metabolism of androgens. It has been demonstrated that alcohol induces the enzyme aromatase that catalyzes the conversion of testosterone in estradiol and androstenedione in estrone [11].

The degree of testicular failure in alcoholic men seems to be related also to the extent of liver damage. A study evaluated testosterone and estradiol levels in patients subdivided into three groups according to the histological severity of liver damage: fatty change, hepatitis, and cirrhosis. Median estradiol levels were above the normal range in males of all three histological categories. Median testosterone concentrations were below the normal range in men with hepatitis and cirrhosis but not in those with fatty liver. Testosterone/SHBG ratio was reduced in patients with cirrhosis. Furthermore, estradiol and testosterone concentrations showed a negative correlation with serum albumin [12].

Although chronic alcohol abuse causes hypogonadism mainly through testicular damage, as evidenced by the high levels of gonadotropins found in most alcoholic men, alcohol is also able to act on the hypothalamus-pituitary axis and its effects at central level are more evident during acute ingestion. Ida and colleagues studied the effect of acute and repeated alcohol administration on plasma prolactin, luteinizing hormone (LH) and testosterone on adult healthy male volunteers [13]. They found that prolactin increased and testosterone decreased after thirty minutes from acute alcohol ingestion, but returned quite rapidly at baseline levels, while LH levels did not change significantly. The repeated alcohol ingestion over seven consecutive evenings did not lead to the development of tolerance to these hormonal changes. The Author hypothesized that alcohol might inhibit the release of hypothalamic dopamine to the hypophyseal-portal system and that hyperprolactinemia could be partially responsible for testosterone decrease after acute alcohol assumption [13]. Otherwise, in chronic abuse, alcohol seems not to affect prolactin levels [9].

A direct action of alcohol at pituitary level, resulting in the inhibition of LH release, has also been hypothesized. In animal models, the suppression of β-LH gene expression and protein release from the pituitary gland after ethanol exposure has been demonstrated [14]. Furthermore, alcohol is able to increase dose-dependently β-endorphin-like peptides release from the hypothalamus. β-endorphin can, in turn, suppress the production and release of gonadotropin-releasing hormone (GnRH) at neuronal level, and of testosterone from the testis [15]. Finally, the high estrogen levels found in alcoholic men can exert negative feedback on gonadotropin release contributing to further reduce testosterone production with a central mechanism [14].

The increase in estrogens concentration due to the enhanced testosterone and androstenedione aromatization and to the altered estrogens’ breakdown by the damaged liver, together with the reduction in testosterone production, is responsible for the gynecomastia that frequently occurs in alcoholic cirrhotic patients [14].

### 2.2. Effects on Spermatogenesis

In the 1980s, Van Thiel and colleagues obtained testicular histology from five chronic alcoholic men and observed a profile characterized by loss of germ cells, peritubular fibrosis and collapse, and aggregate of residual Leydig cells between the abnormal seminiferous tubules [8]. They found the same histologic profile in alcohol-fed rats [8].

Ten years later, prospective autopsy studies showed that spermatogenic arrest and Sertoli cells-only syndrome are present in, respectively, 50% and 10% of heavy drinkers, while less than 20% of non-alcoholic controls have alterations of spermatogenesis [16]. The testicular damage was not related to the extent of liver damage because most of the men with Sertoli cells-only syndrome had not cirrhosis, but it was strongly correlated with daily alcohol intake: testicular alterations were more likely to be present in men who drank more than 80 g of alcohol daily [17].

More recent studies demonstrated that the reduction of testicular germ cells in alcoholic men is due to the activation of apoptosis. In the mouse testis, ethanol-induced apoptosis through the increased expression of Fas/Fas-L and p53, the up-regulation of Bax/Bcl-2 ratio, and the activation of caspase-3 [10].

The testicular histological alterations found in heavy drinkers clinically translate in a significant reduction in sperm, up to azoospermia [9]. The spermatogenetic damage caused by alcohol abuse seems to be reversible. Case reports and animal studies showed that spontaneous recovery of spermatogenesis could occur starting from 10–12 weeks after alcohol consumption withdrawal [18,19,20,21].

Unlike alcohol abuse, a moderate alcohol intake seems not associated with altered sperm concentration [22]. A dose-dependent effect was demonstrated by Jensen and colleagues, who found a progressive deterioration of sperm count, concentration and morphology with the increase in the amount of alcohol consumed, more evident in patients with a weekly alcohol intake higher than 25 units [23]. This trend has been recently confirmed: Boeri and colleagues showed that heavy drinking was associated with a lower sperm concentration than moderate drinking and/or abstaining. They also reported that drinking and smoking concomitantly has an even greater detrimental effect on semen parameters [24]. A recent meta-analysis assessed the association between alcohol intake and semen quality, examining the pooled data of fifteen cross-sectional studies, with a total of 16,395 enrolled men. Results showed a detrimental effect of alcohol on semen volume and sperm morphology but not on sperm concentration [25]. However, most of the studies included in the analysis evaluated men with moderate alcohol intake or did not report the exact alcohol intake of the patients. Anyway, results confirm that occasional alcohol consumption did not adversely affect semen parameters [25]. Some evidence indicate even a positive effect of moderate alcohol intake on semen quality [26].

It has been demonstrated that the homozygous deletion of the glutathione S-transferase *(GST)-M1* gene increases the susceptibility to develop alcoholic liver cirrhosis in response to the toxic effects of alcohol chronic abuse [27]. Similarly, the association between alcohol-induced alteration of human spermatogenesis and GST-M1 genotype has been investigated. An autopsy study revealed that heavy drinkers with GST-M1 ’null’ genotype developed less frequently disorders of spermatogenesis; so, Authors hypothesized the GST M1 locus may be associated with susceptibility to alcohol-induced testicular damage [28].

A state of protein malnutrition, with nutritional imbalance or deficiencies, could contribute to the onset of spermatogenesis disorders, as well as testosterone deficiency, in alcoholic men. Due to low dietary intake or excessive loss (i.e., by vomiting or diarrhea), alcoholic patients can suffer from the lack of some minerals and micronutrients such as zinc, magnesium, folate, and vitamins (A, D, E), belonging to the human not enzymatic antioxidant system. The lowering of antioxidant defenses exposes the germ cells to the deleterious effect of oxidative stress [29].

## 3. Cigarette Smoking

Even more than alcohol, cigarette smoking is recognized as a risk factor for many diseases: cardiovascular diseases, lung diseases, malignant neoplasms, etc. Since the 1980s, scientific literature has been interested in evaluating the effects of cigarette smoking on reproductive function. Tobacco smoke is a complex mixture of over than 8,700 substances. Harmful cigarette smoke constituents include carbon monoxide, nitrogen oxide, ammonia, heavy metals, various polycyclic aromatic hydrocarbons and aldehydes, such as hydroquinone, catechol, acrolein, crotonaldehyde, and formaldehyde [30]. Recently, even nicotine, the major psychoactive substance in cigarette smoke, has been called into question in the pathogenesis of smokers sperm alterations with a possible neuroendocrine mechanism [31]. Indeed, it has been demonstrated that nicotine and its metabolites are capable to cross the blood-testis barrier [32].

### 3.1. Effects on Testosterone Production

In vivo animal studies demonstrated that rat exposed to cigarette smoke show low testosterone elevation in response to human chorionic gonadotropin (hCG) stimulation compared to non-exposed rats [33]. The chronic administration of nicotine in male rats determined a reduction in testosterone and estradiol levels. This effect was counteracted by mecamylamine, an inhibitor of nicotine, proving that nicotine has a specific gonadotoxic effect [34]. The decreased testosterone levels seem to turn to normality after nicotine cessation, indicating a potential reversible effect of nicotine on Leydig cells function [35]. It has been demonstrated that nicotine and its metabolites inhibit multiple steps in testosterone biosynthesis. In rat Leydig cells, nicotine and cotinine produce a dose-dependent increase in progesterone levels and a dose-dependent decrease in testosterone concentration, by the inhibition of 17α-hydroxylase, 17,20-lyase, and 17-ketosteroid reductase [36]. Furthermore, nicotine exerts cytotoxic effects on mouse Leydig cells in a concentration- and time-dependent manner inducing apoptosis, as demonstrating by the increase in Bax (a pro-apoptotic protein) and caspase-3 expression and a decrease in Bcl-2 (an anti-apoptotic protein) expression [37].

Despite the evidence in animal studies of the cytotoxic effect of nicotine and cigarette smoke on Leydig cells, in male smokers a reduction of testosterone levels has not been clearly demonstrated. Conversely, most of the studies found higher testosterone levels in smokers than in non-smokers [38,39,40,41,42]. Higher testosterone and LH levels with higher LH/free testosterone ratio were found with increased smoking, hence authors hypothesized a compensated Leydig cell failure in smokers [40]. The concurrent increase in testosterone and LH levels was confirmed for current smokers of five or more cigarettes/day but not for smokers of less than five cigarettes/day [41]. Recently, a meta-analysis of 22 studies with a total of 13,317 men, confirmed that smokers show testosterone levels higher than non-smokers [43].

Since higher testosterone concentration should lower LH levels through the negative feedback exerted at central level, it has been postulated that tobacco alters hypothalamus-pituitary axis enhancing GnRH or LH release. Alternatively, it has been hypothesized that higher testosterone levels could be the basis of the smoking habit, favoring addiction to smoking, since men with higher testosterone levels tend to adopt more risky lifestyles [38,41]. Another mechanism that could contribute to testosterone rise in smokers is the inhibition of testosterone breakdown by nicotine’s metabolites. Indeed, the uridine 5′-diphospho (UDP)-glucuronosyltransferase (UGT) enzyme superfamily catalyzes the glucuronidation of both testosterone and nicotine’s metabolites; so, cotinine and trans-3′-hydroxycotinine can compete with testosterone for binding to the catalytic site and prevent androgen inactivation [43].

Nicotine at the mesolimbic level increases dopamine release, which in turn inhibits prolactin release from the hypophysis. Some studies showed lower prolactin levels in smokers than in non-smokers [38]; however, other studies reported higher prolactin levels in smokers [41]. This paradoxical effect could be explained by the action of endogenous opioids, released in response to nicotine, which could reduce dopamine release.

### 3.2. Effects on Spermatogenesis

The association between cigarette smoking and sperm concentration has been studied from the 80s. A first meta-analysis conducted in 1994 showed that smokers had a sperm density 13–17% lower than that of nonsmokers [44]. Subsequent studies confirmed this finding. Ten years later Künzle and colleagues found in smokers a decrease of about 15% in sperm density and of about 17% in total sperm count compared to nonsmokers [45]. Another study on over 2,500 men found an inverse dose–response relation between smoking and sperm count, with a 19% lower sperm concentration and a 29% lower total sperm count in heavy smokers compared to non-smokers [40]. Other studies failed to demonstrate a detrimental effect of cigarette smoking on sperm concentration [46,47] or found a trend for a reduction of sperm count with the increasing number of smoked cigarettes, without reaching statistical significance [48]. However, a more recent meta-analysis with a wider sample size confirmed that smoking is a risk factor for all sperm parameters, including sperm density and total count, in both infertile and healthy men [49]. Another meta-analysis was conducted in 2016, including only studies performed after the introduction of 2010 WHO manual for the laboratory evaluation of human semen. This meta-analysis involved a total of almost 6,000 participants, and found a reduced sperm count in smokers, with higher effect size in infertile men and in moderate/heavy smokers [50]. The latest meta-analysis, performed in 2019, evaluated sixteen studies including almost 11,000 infertile male participants subdivided in smokers and non-smokers and found that oligozoospermia is significantly more frequent in smokers with a relative risk of 1.29 [51].

It has been demonstrated in vitro that spermatozoa from healthy, non-smoker men incubated with cigarette smoke extract show an increase in phosphatidylserine externalization, an early apoptotic sign, and in DNA fragmentation, a late apoptotic sign, in a concentration- and time-dependent manner [52]. Similarly, the incubation of spermatozoa with increasing concentration of nicotine reduces the percentage of viable spermatozoa and increases the number of spermatozoa with altered chromatin compactness, or DNA fragmentation [53]. This finding indicates that nicotine itself is, at least in part, responsible for the detrimental effect of cigarette smoke on sperm. Indeed, nicotine and its major metabolites, cotinine and trans-3’-hydroxycotinine, are capable to cross the blood-testis barrier and their concentrations are similar or even higher in the seminal plasma than in the blood of smokers [32].

Nicotine binds to a class of ionotropic acetylcholine receptors, the nicotinic receptors (nAChR), made up of five subunits, whose expression has been demonstrated also on human spermatozoa. Since hexamethonium, the main antagonist for the neuronal nAChR, is able to reverse the detrimental effects of nicotine on non-conventional sperm parameters, including apoptotic signs, a possible neuroendocrine mechanism has been postulated for nicotine-related sperm damage [31]. In semen of non-smokers, only a homomeric nicotinic receptor consisting of five α7 subunits is present; however, it has been demonstrated that cigarette smoking may stimulate the expression of some other subunits, such as α9 subunit, that in pregnant smoker women is involved in the vasoconstriction, disepithelization, and apoptosis of the placenta. The aberrant expression of subunit different from α7 in semen could contribute to the functional alterations found in the spermatozoa of smokers and could represent a marker for smoking-related sperm damage [54].

In rats exposed to nicotine ultrastructural alterations of germ cells, peritubular structures, and Sertoli cells were found. These alterations include: thickening of the tunica propria; degeneration of junction between the Sertoli cells; irregular cristae and electron-dense matrix inside Sertoli cells’ mitochondria; spermatids with altered cytoplasm-nucleus ratio, cytoplasmic electron-dense lipid droplets, and abnormal acrosome [55].

In addition to the gonadotoxic action of nicotine, reactive oxygen species (ROS) have been called into question as determinants of sperm damage in smokers. Indeed, smokers, have higher malondialdehyde, and protein carbonyls level (markers of oxidative stress), and lower glutathione, ascorbic acid, zinc, superoxide dismutase, catalase, and glutathione-S-Transferase than non-smokers [56]. Cadmium and lead levels in seminal plasma of smokers were correlated with the oxidative stress markers [57]. Erythroid 2-related factor 2 (NRF2), an antioxidative transcription activator that binds to antioxidant response elements in the promoter regions of target genes, could be implicated in the susceptibility of sperm damage induced by ROS in smokers. Indeed, the NRF2 rs6721961 TT genotype was found to be associated with lower sperm concentrations and sperm counts in heavy smokers [58].

## 4. Caffeine

Caffeine is a methylated xanthine, structurally similar to purine and uric acid, present in coffee, tea, soft drinks, and chocolate. One cup of coffee contains about 137 mg of caffeine, a cup of tea 47 mg, a bottle or can of caffeinated soft drinks (i.e., cola) about 46 mg, a serving of chocolate about 7 mg [59]. Caffeine stimulates heart contraction and rate, dilates blood vessels by relaxing smooth muscles, increases the secretion of catecholamine, increases diuresis, enhances alertness and decreases drowsiness and fatigue. It is able to cross the blood-testicular barrier and it is found in the same concentrations in blood and semen [60]. For these reasons, its effects on reproductive function have been investigated.

### 4.1. Effects on Testosterone Production

In a study of more than 2,500 men, no statistically significant association was found between caffeine intake and reproductive hormones [61]. However, in other studies caffeine consumption was positively associated with total [39,62] or bioavailable testosterone [63].

In animal models, caffeine induces a stress-like hormonal response. The acute treatment with high doses of caffeine caused in male rats a rise above control values in plasma concentrations of corticosterone, progesterone, testosterone, and noradrenaline [64]. Prolonged treatment for up to 30 days determined a significant increase in serum testosterone and a significant decrease in serum luteinizing hormone and follicle-stimulating hormone (FSH) levels [65].

### 4.2. Effects on Spermatogenesis

The evidence about a putative detrimental effect of caffeine on sperm production is controversial. Most of the studies showed no significant differences in sperm concentration in relation to caffeine intake [59,61,62,66,67,68]. Only cola soft drinks consumption has been associated with lower sperm count and concentration, but this effect is not directly related to caffeine because its content is much lower than in coffee [61]. However, an intake of more than 700 mg/d caffeine has been associated with a decreased fecundability rate both in men than in women [67], and a paternal intake of more than 272 mg/d has been associated with a decreased live birth rate in intracytoplasmic sperm injection (ICSI) cycles [59].

A recent systematic review of 28 studies and about 20,000 men confirmed that caffeine intake does not influence sperm count and concentration. However, data meta-analysis was not performed because of the extreme heterogeneity in exposure measurement (i.e., weekly intake of coffee alone or of different sources of caffeine), study design, and studied outcomes [25]. The single existing meta-analysis included only two studies and concluded for no statistically significant effects of coffee consumption on sperm density [49].

An animal study showed that Wistar rats orally treated with caffeine for 30 days have a decreased sperm count which does not ameliorate after caffeine withdrawal. Histological sections of the testis in treated rats showed subcapsular and interstitial congestion [65].

Caffeine acts as an antagonist of adenosine receptors. Adenosine receptors are also present in Sertoli cells where they seem to promotes the production of lactate, the preferred metabolic substrate of germ cells [69]. It has been demonstrated that the incubation of Sertoli cells with low-moderate doses of caffeine enhances lactate production and increases the expression of GLUT1 and GLUT3. However, at high concentration caffeine exerts on Sertoli cells a pro-oxidant effect [69].

## 5. Cannabis

Extract from Cannabis sativa, commonly referred to as marijuana, is the most widely used illegal drug in many countries. The major psychoactive substance contained in cannabis, Δ9-tetrahydrocannabinol (THC), is able to interact with cannabinoid receptors CB1 and CB2, belonging to the superfamily of G-protein coupled receptors. In humans, CB1 is localized in nervous system and other tissues, including reproductive system (ovary, uterine endometrium in women, testis and vas deferens in men), while CB2 is found predominantly in immune cells but also in Sertoli cells [70]. The endogenous ligands for cannabinoid receptors are the endocannabinoids produced and released on-demand by neurons and peripheral cells. The main endocannabinoids are anandamide and 2-arachidonoylglycerol [71]. The endocannabinoid system is involved in the regulation of reproductive function [72]. For these reasons, the effects of cannabis on male reproductive function have been investigated for almost 50 years, leading to contrasting results.

### 5.1. Effects on Testosterone Production

Since 1970s, a dose-related decrease in testosterone levels has been demonstrated in chronic marijuana smokers [73]. In acute administration, plasma LH was significantly depressed and cortisol was significantly elevated after smoking marijuana, indicating that cannabis decreases testosterone levels with a central mechanism [74]. Other studies failed to demonstrate statistically significant differences in plasma testosterone levels between occasional and chronic marijuana smokers [75], between occasional smokers and controls [76], or between daily cannabis users and controls [77]. A recent population study on over 1,500 U.S. men found no differences in serum testosterone levels among ever users of marijuana compared to never users. However, testosterone concentrations were higher in men with more recent marijuana use, especially in men aged 18–29 [78]. A study on over 1,200 young healthy men reported, similarly to tobacco smokers, an increase in testosterone levels in marijuana smokers [79]. A recent systematic review on 15 clinical studies and 21 animal/in vitro studies concluded for a not significant relationship between long-term cannabis consumption and alteration of the hypothalamic–pituitary–testicular axis hormones [80].

Despite these conflicting data, an inhibitory action of THC at central level is plausible since it has been demonstrated that CB1 receptors are present both at pituitary (expecially in lactotrophs and gonadotrophs) and hypothalamic (in GnRH neurons) level. Endocannabinoids depress the pituitary secretion of thyroid-stimulating hormone (TSH), LH, growth hormone (GH), and prolactin, and the hypothalamic GnRH release in rats [81]. Specifically, the CB1 receptor agonist anandamide suppresses LH and testosterone secretion [82]. In vitro, it has been demonstrated that endocannabinoids inhibit gamma-aminobutyric acid (GABA) A receptors drive in GnRH neurons, determining a decrease in GnRH neuron firing rate [83].

However, a direct effect of THC in testis has also been demonstrated in animal models. In rats, acute and chronic administration of THC significantly depressed testosterone formation in testis microsomes [84]. In in vitro studies, murine Leydig cells incubated with THC produced less testosterone in response to hCG and dibutyryl-cAMP [85]. Furthermore, a reduced expression of LH receptor on testis and a reduced activity of testicular 3β-hydroxysteroid dehydrogenase has been demonstrated in mice fed with a preparation containing cannabis [86].

### 5.2. Effects on Spermatogenesis

Parallel to the reduction of testosterone levels, a higher prevalence of oligozoospermia has been found in marijuana smokers since the 70s. Kolodny and colleagues found oligozoospermia in 35% of men who used marihuana without other drugs at least four days a week for a minimum of six months [73]. In a study on more than 1,200 healthy young men, the Authors found a lower sperm concentration and a lower total sperm count among men smoking marijuana more than once per week. The concomitant use of other recreational drugs was associated with a further worsening of sperm production [79]. Other studies failed to demonstrate a decrease in sperm count in marijuana smokers [87]. A recent systematic review of seven clinical studies and 23 animal/in vitro studies concluded that cannabis consumption exerts a negative impact on semen parameters, but it considered also the alteration of sperm motility and morphology that is not an argument of our review [80].

It has been demonstrated that endocannabinoids at male reproductive level inhibit acrosome reaction and sperm capacitation, and induce programmed cell death in Sertoli cells (effect counteracted by FSH) [71]. Since clinical studies on humans are not feasible for ethical reasons, most of the studies on spermatogenesis were carried on animals and demonstrated a detrimental effect of THC on germ cells. For example, daily administration of cannabis extract in dogs for one month produced a complete arrest of spermatogenesis, with extensive fibrosis and exfoliation of the seminiferous elements [88]. Regressive changes have also been demonstrated in the testes of mice fed with bhang, an Indian edible preparation of cannabis, for over 30 days [86]. The chronic administration of HU210, a synthetic analogue of THC and potent agonist of CB1 and CB2 receptors, determined in male rats a significant reduction in sperm count and daily sperm production, and a reduction in the number of Sertoli cells. No significant differences were observed after acute treatment [89]. Conversely, a recent study conducted on mice treated with a daily dose of 10 mg/kg THC for 30 days found no changes in testis weight nor alterations in spermatogenesis and sperm concentration. Moreover, the morphology of Sertoli and Leydig cells was normal, and apoptosis - evaluated with TUNEL test - was not different between the two groups [90].

## 6. Cocaine

Cocaine is an alkaloid obtained from the leaves of many species of the Erythroxylaceae family. It is a powerful local anesthetic, it alters thermoregulation and exerts stimulating effects on cardiovascular system, central and peripheral nervous system [91]. Cocaine abuse and dependence are very frequent, especially in Western countries. However, the relationship between cocaine intake and male reproductive function has not been extensively studied. In infertile male population, a low prevalence of cocaine use (˂1%) has been reported, but men consuming cocaine are more likely to use other illicit drugs and substances (e.g., alcohol, tobacco) which may negatively impact their reproductive function [92].

### 6.1. Effects on Testosterone Production

Increase in LH, adrenocorticotropin hormone (ACTH), and cortisol, decrease in prolactin e unchanged testosterone levels have been found in men after intravenous low-dose cocaine injection (0.2–0.4 mg/kg) [93,94]. Conversely, in men who chronically use cocaine, lower free testosterone concentrations have been found, while gonadotropins did not differ compared to nonusers [95]. Furthermore, some cases of cocaine-induced panhypopituitarism have been described: intranasal cocaine abuse can cause pituitary infarction [96] or the production of human neutrophil elastase-anti-neutrophil cytoplasmic antibodies (HNE-ANCA), leading to a pituitary inflammatory process [97].

Regarding evidence on animals, a biphasic effect on the testosterone concentration has been found in rats following intraperitoneal injections of high doses of cocaine: testosterone initially raised and then dropped quickly [98]. In another study, male Wistar rats treated with intraperitoneal injections of low doses of cocaine showed an increase in testosterone concentration and unaltered LH levels. The same effect was not demonstrated with high doses of cocaine [91]. Similarly, the chronic administration of cocaine (15 mg/kg for 100 days) in rats did not induce changes in testosterone, FSH and LH levels [99]. In male Rhesus monkeys, a single injection of low dose cocaine caused a rapid increase in LH levels, while testosterone concentration did not change. The Authors hypothesized that LH release following cocaine injection was due to a burst of hypothalamic GnRH [100].

### 6.2. Effects on Spermatogenesis

Since the 1990s, the association between oligozoospermia and cocaine consumption has been investigated. Bracken and colleagues found an odds ratio of 2.1 of having sperm counts <20 × 106 mL in men who referred cocaine use [101].

In male rats repeated intraperitoneal injections of low and high doses of cocaine produced a decline in the number of normal seminiferous tubules of respectively 50% and 40%, and an increase in regressive tubules of 50% and 60%. Testis weight was not significantly reduced after treatment; however, the testicular volume decreased after high doses of cocaine. The main cytological changes found in spermatogonia, spermatids, and Sertoli cells were lipid droplets, vacuoles and giant mitochondria [91]. Effects of long-term cocaine administration on spermatogenesis were similar: peripubertal rats treated for 100 days with cocaine showed a reduced mean diameter of seminiferous tubules and thickness of the germinal epithelium. The number of spermatids also decreased. These findings indicate a significant toxic action on spermatogenesis, that Authors attributed to the ischemic effect of cocaine. Indeed, cocaine, enhancing norepinephrine and epinephrine release, induces intense vasoconstriction [99]. It has been demonstrated that cocaine chronic administration increases germ cell apoptosis, evaluated by TUNEL assay, of about 25%. Since ROS activate apoptosis and are generated during the metabolization of cocaine and during reperfusion injury, they have been implicated in the cocaine-induced programmed cell death [102,103].

## 7. Amphetamine, Methamphetamine, and MDMA (Ecstasy)

Amphetamine is a drug derived from phenethylamine with the addition of an α-methyl group that protects it against metabolism by monoamine oxidase. It stimulates the release of monoamines (i.e., dopamine and noradrenaline) in central nervous system and it has been used since the 1930s in the treatment of psychiatric disorders [104]. It is still employed in the U.S. and in some European countries in the management of Attention Deficit/Hyperactivity Disorder (ADHD); however, for its euphoric and stimulant effects, it is also taken for recreational purposes. Methamphetamine is obtained from the methylation of amphetamine which confers it greater psychostimulant activity. It is also known with the street names Speed and Meth Crystal and it is often abused for recreational purpose. 3,4-Methylenedioxymethamphetamine (MDMA), commonly known as Ecstasy, is a synthetic amphetamine, with serotonin and dopamine-releasing properties. It has several stimulating and inhibiting effects on central and peripheral nervous system (i.e., euphoria, increased energy, insomnia, enhanced sensory perception; attention and memory deficit, reduction of psychomotor speed and executive cognitive function); furthermore, it alters circadian rhythms and thermoregulation, causing hyperthermia. It is the entactogen molecule par excellence since it is able to produce feelings of empathy [105]. Amphetamines have demonstrated, in experimental settings, several effects on testicular function, as summarized below. However, studies on humans are not available.

### 7.1. Effects on Testosterone Production

In vivo and in vitro studies demonstrated that amphetamine decreases testosterone production and increases the generation of testicular cyclic AMP in rats. Also hCG-stimulated testosterone release was reduced in rats following a single intravenous injection of amphetamine, while plasma LH levels did not change. Authors hypothesized that amphetamine could act directly and dose-dependently on Leydig cells, and that the activation of adenylate cyclase could be responsible for the inhibition of testosterone production after amphetamine administration [106]. The same Authors demonstrated subsequently that in Leydig cells amphetamine decreases the activity of the steroidogenic enzymes 3b-hydroxysteroid dehydrogenase, P450c17, and 17-ketosteroid reductase, and attenuates Ca^2+^ influx through L-type Ca^2+^ channel [107]. Another study evaluated the effects of amphetamine on steroidogenesis in MA-10 mouse Leydig tumor cells, which produce progesterone as the major steroid instead of testosterone in response to hCG. Contrary to previous studies, the Authors demonstrated a stimulatory action of amphetamine, which directly enhanced hCG-induced progesterone production in cells by increasing the activity of the P450scc enzyme. No effects on the activity enzymes 3b-hydroxysteroid dehydrogenase was found [108].

In an in vivo study, a single intraperitoneal administration of methamphetamine exerted a biphasic effect on testosterone production in mice: serum testosterone concentrations initially decreased and then increased, reaching a level higher than basal after 48 hours. The Author postulated that, similarly to amphetamine, methamphetamine could decrease testosterone production acting at testicular level [109]. In another study, rats chronically treated with high doses of methamphetamine exhibited lower testosterone levels compared to controls [110]. An increase in testicular GABA concentration has also been reported in methamphetamine treated rats [111]. Since GABA is involved in the proliferation of Leydig cells and testosterone production, Authors hypothesized that the increase in GABA concentration could represent a compensatory response to the detrimental effects of methamphetamine on Leydig cells [111].

It has been demonstrated that MDMA suppresses the hypothalamic-pituitary-gonadal axis in male rats. Following acute or chronic MDMA administration, adult male Sprague-Dawley rats showed lower expression of GnRH mRNA and decreased serum testosterone concentrations compared to controls. LH, progesterone, and estradiol concentrations were not affected, suggesting a diminished drive from hypothalamic GnRH neurons as the cause of the hypothalamic-pituitary-gonadal axis inhibition [112]. Since both dopamine and serotonin receptors are expressed in the preoptic area, where GnRH cell bodies allocate, these two neurotransmitters are probably implicated in the inhibition of GnRH mRNA expression [112]. Conversely, another study, in which MDMA was administered to male rats subcutaneously during 12 weeks one a day for three consecutive days a week, simulating human weekend associated consumption, failed to demonstrate any effects of MDMA on the hypothalamus-pituitary-gonadal axis hormones [113].

### 7.2. Effects on Spermatogenesis

In has been demonstrated that methamphetamine induces apoptosis in murine testicular germ cells: in mice treated with increasing intra-peritoneal doses of methamphetamine a dose-dependent percentage rise of the apoptotic tubules was detected by TUNEL. Histological changes found in the murine testis include vacuolization of spermatogonia and derangement of cell layers [114]. These findings were confirmed by another study demonstrating that methamphetamine administration decreases significantly cell proliferation and increases apoptosis in both rats spermatogonia and primary spermatocytes, altering proliferation/apoptosis ratio [115]. Another confirmation came from the study of Nudmamud-Thanoi and Thanoi, were rats treated acutely and sub-acutely with methamphetamine showed an increase in TUNEL-positive cells in seminiferous tubules and a parallel reduction in epididymal sperm count [116]. In a more recent study, rats receiving 5 ml/kg intraperitoneal methamphetamine for 7 and 14 days showed a significant decrease in the number of spermatogonia, primary and secondary spermatocytes, and in spermatogenesis indices (tubular differentiation index, spermiogenesis index, repopulation index, and mean seminiferous tubules diameter) compared with controls [117].

Apoptosis in germ cells could be induced by hydroxyl radical formation, overproduction of serotonin or methamphetamine-induced testicular thermic rise [115]. In rats chronically treated with high doses of methamphetamine a significant decrease in GSH/GSSG ratio was recorded, indicating oxidative stress. Antioxidant enzymes (superoxide dismutase, catalase, and glutathione peroxidase) initially declined and then returned to normal, suggesting an adaptive response to scavenge ROS produced during methamphetamine metabolism. In parallel, a decreased expression of Bcl2 and increased levels of cleaved caspase-3 were found, indicating the activation of apoptosis. The Authors also reported a reduced epididymal sperm count in treated rats [110]. Methamphetamine seems also to reduce in male rats the expression of progesterone and estrogen receptors [116]. These receptors are normally expressed both in germ cells and in Sertoli cells and seem to play a role in germ cells proliferation and differentiation during development, inhibition of apoptosis, spermiogenesis, and sperm capacitation [118]. Finally, increased GABA concentrations have been found in rat testicular tissue after methamphetamine administration. Since GABA activity can suppress the proliferation of spermatogonial stem cells, the increase in its concentration could lead to alteration of proliferation/apoptosis ratio [111].

Also MDMA is able to cause histological alterations in rat testis. After subcutaneously MDMA administrations one a day for three consecutive days a week over a 12 weeks period, mild tubular degeneration and interstitial edema were observed in rat testicular tissue. The Comet assay showed a significant dose-related increase in sperm DNA damage; however, surprisingly, there was a significant enhancement of sperm count and a decrease in spermatids count in treated rats [113]. In another study conducted on rats, MDMA significantly increased the number of apoptotic TUNEL-positive cells in both germinal epithelium and Leydig cells, while germinal epithelium thickness and diameter of the tubules decreased. In parallel, an increase in body temperature and in immunoreactivity of heat shock protein 70 (HSP70) was observed [119]. Since testes are sensitive to temperature and HSP are produced in response to thermic stress to stop caspase activation and inhibit the apoptotic process, the Authors hypothesized that MDMA-induced hyperthermia could activate apoptosis in rat testicular tissue [119].

## 8. Opioids

Opioids are derived from opium, extracted from the seed pod of the opium poppy (Papaver somniferum). They include opium alkaloids, such as morphine, and synthetic derivatives, such as codeine and heroin. They have the ability to bind with three main classes of receptors - mu (µ), delta (δ), and kappa (κ) - belonging to the superfamily of G-protein coupled receptors, which usually interact with three major classes of endogenous opioid peptides (endorphins, enkephalins, and dynorphins) [120]. The endogenous opioids are physiologically implicated in the regulation of several functions: motor, immune, gastrointestinal, cardiovascular, neuroendocrine, cognitive, and, more notoriously, nociceptive function. Opioid analgesics, such as oxycodone, hydrocodone, propoxyphen, fentanyl, and methadone are frequently prescribed in muscular-skeletal and rheumatological conditions because they are effective in reducing pain, inexpensive and long-lasting. However, their potential risk for addiction is well known [120]. Opioid receptor antagonists, such as naloxone and naltrexone, are clinically used to reverse the effects of opioid overdose. The inhibitory effects of opioids on testosterone production are widely known; while effects on spermatogenesis are still controversial.

### 8.1. Effects on Testosterone Production

Opioid-induced androgen deficiency (OPIAD) is nowadays a recognized syndrome characterized by decreased levels of testosterone, reduced libido and muscle mass, fatigue, and osteopenia [121]. A cross-sectional study examined the prevalence of opioid use in the last 30 days in almost 5000 men and women aged 17 years and older from the general population. Testosterone levels of opioid-exposed were compared with those of non-exposed subjects. The Author found an overall OR = 1.40 of having low testosterone levels in opioid-exposed subjects. The most commonly used opioids were hydrocodone, oxycodone, and tramadol [122].

The inhibitory effects of opioid drugs on hypothalamic–pituitary–testicular axis have been known for over 40 years [123]. Endogenous and exogenous opioids inhibit hypothalamic GnRH secretion, disrupting its normal pulsatility and leading to decreased LH levels. As a consequence of reduced GnRH secretion, LH and therefore testosterone levels decrease, determining hypogonadotropic hypogonadism. The FSH levels would seem to be less affected by opioids administration, in both animals and men [120]. Hyperprolactinemia may contribute to opioids central inhibitory activity. Indeed, acute administration of opioids increases prolactin levels in both human and animal models, while their chronic administration could have variable effects on prolactin secretion based on the type of opioid used [120].

Men treated with long-term intrathecal opioid administration exhibit lower LH, serum testosterone and free androgen index than controls [124]. Low levels of total and free testosterone and high levels of SHBG have been also reported in heroin-addicted men [125,126] and in men consuming sustained-action oral opioids for control of non-malignant pain [127]. In a study on opium-addicted Iranian men, mean serum levels for LH, total, and free testosterone were significantly lower than in controls [128].

It has been hypothesized that not all opioids exert the same effect on testosterone production. Indeed, buprenorphine, a partial µ-opioid receptor agonist and pure antagonist at the κ-opioid receptor, used to treat opioid dependence, affects fewer testosterone levels than methadone [129,130,131]. Contrary to this hypothesis, a meta-analysis including 12 studies on men found a difference in testosterone levels of 165 ng/dL in men using opioids compared to controls, with no significant differences between the different types of opioids. The Authors concluded that all opioids suppress testosterone; however, only one study on buprenorphine was included in the meta-analysis [132]. It has also been demonstrated that long-acting opioids induce more frequently hypogonadism than short-acting opioids, probably for their prolonged inhibitory action on GnRH release [133].

Several studies demonstrated the inhibitory effects of opioids on the hypothalamic-pituitary-gonadal axis also in animal models. Four days of treatment with morphine determined in rat hypothalamus a marked decrease in GnRH mRNA levels, indicating that opioids inhibit the biosynthesis of the neuropeptide [134]. In vivo and in vitro studies demonstrated that methadone inhibits the dopamine-stimulated release of GnRH in rats [135]. Following morphine injection, testosterone levels decreased in rats dose-dependently and this effect was counteracted by the opioid antagonists naltrexone and naloxone [136]. Rats intraperitoneally injected for 30 days with tramadol hydrochloride, a widely used analgesic drug which binds to µ-opioid receptors more weakly than morphine, exhibited lower LH, FSH and testosterone levels than controls after treatment [137]. Another study found reduced LH levels in rats chronically treated with morphine, but FSH and testosterone levels did not significantly differ compared to controls. However, in this study the morphine administered dose was relatively low [138].

### 8.2. Effects on Spermatogenesis

It has been demonstrated that human spermatozoa express µ-, δ-, and κ- opioid receptors, located in the head, in the middle region, and in the tail of the sperm [139]. In a study conducted in Iran, where opium consumption is relatively common in the male population, opium-addicted men showed more frequently oligozoospermia than controls. Furthermore, they exhibited lower antioxidant activity and higher sperm DNA fragmentation index compared to healthy age-matched male volunteers [128].

Rats chronically treated with tramadol exhibited histological degenerative changes in the seminiferous tubules, in Sertoli cells and in Leydig cells: tubules showed a decrease in mean diameter and epithelial height, shrinkage, separation of tubular basement membrane, and disorganization and vacuolization of spermatogenic layers; Sertoli cell presented vacuolation, huge lipid droplets and disrupted junctions; Leydig cells had euchromatic nuclei and dilated smooth endoplasmic reticulum [137]. The Authors suggested that regressive histological changes following tramadol administration were linked to opioid-induced testosterone deprivation [137].

Recently, increased caspase-3 and decreased anti-apoptotic protein Bcl-2 expression have been described in rats chronically treated with tramadol. Authors hypothesized that the apoptotic process was induced by oxidative stress since malonldialdehyde levels were increased, while the antioxidant enzymes activity was decreased in treated rats [140]. Interestingly, tramadol withdrawal improved but did not normalize apoptotic changes in testicular tissues, indicating permanent opioid-induced damage [140].

Another study failed to demonstrate a reduction of seminiferous tubules diameter in rats chronically treated with morphine, even if sperm count was three times lower in treated rats than controls. Also the distribution of Leydig and Sertoli cells did not differ from the two groups. However, the dose of morphine administered was on average lower than in the other studies [138]. These findings are partially in agreement with a study of Cicero and colleagues, where morphine treatment did not induce histological changes in the testis of rats treated with morphine. However, in this study – were morphine was administered at higher dosage but only for 14 days – also sperm count was normal [141]. These results could indicate that morphine exerts less gonadotoxic effects than other synthetic opioids, or that high dose and duration of treatment are both needed to reveal the detrimental effects of morphine on testis histology and function.

## 9. Anabolic-Androgenic Steroids

Anabolic-androgenic steroids (AAS) are the most used drugs by athletes, amateur sportsmen, and body-builders all over the world to improve sports performance and/or physical appearance. The global lifetime prevalence rate of their use is 6.4% for males [142]. At least 30 different AAS exist, including testosterone, its 17α-alkyl-derivates (e.g., oxandrolone, stanozolol), its 17β-ester-derivates (e.g., nandrolone, testosterone esters), and its precursors (androstenedione, dehydroepiandrosterone) [143]. The detrimental effect of AAS on endogenous androgens production and spermatogenesis are widely known and below summarized.

### 9.1. Effects on Testosterone Production

The AAS-induced hypogonadism is a widespread phenomenon. Indeed, in men younger than 50 years, the most common etiology of profound hypogonadism, defined as testosterone 50 ng/dl or less, is just the previous assumption of AAS [144]. Anabolic-androgenic steroids suppress gonadotropin release from the pituitary gland by a negative feedback mechanism, exerted on both pituitary gland and hypothalamic GnRH-releasing cells. This results in a down-regulation of gonadotropins and a decreased secretion of endogenous steroids [143]. Therefore, while during administration abusers may exhibit high androgen levels, they became hypogonadal following AAS discontinuation, especially after prolonged use.

A recent systematic review and meta-analysis revealed that long-term AAS use results in prolonged hypogonadotropic hypogonadism. In almost all studies included in the meta-analysis, LH and FSH serum levels decreased during AAS use and progressively increased, until returning to the basal levels, following AAS withdrawal. Conversely, testosterone blood concentration, which decreased during AAS abuse, remained lower compared to baseline following 16 weeks of AAS discontinuation [145].

The recovery of hypothalamus-pituitary-testicular axis can take from a few weeks to over a year after AAS withdrawal [146,147]. However, a recent cross-sectional case-control study, involving 37 current AAS abusers, 33 former AAS abusers who ceased steroids for at least 1.7 years, and 30 healthy controls, found that almost one-third of former AAS abusers had total testosterone levels below 12.1 nmol/L, indicating persistent hypogonadism [148]. Indeed, it has been demonstrated in animal studies that a permanent depletion of Leydig cells may occur following AAS administration. A reduction of interstitial cells’ number has been reported in testicular tissue of pre-pubertal and adult male rats treated with high doses of AAS. Differently from pre-pubertal animals, which obtained a complete recovery of the Leydig cells number following AAS withdrawal, in adult rats Leydig cells did not return to the control level, suggesting a long-lasting alteration [149].

### 9.2. Effects on Spermatogenesis

The AAS abuse leads to severe oligozoospermia up to azoospermia because the elevated levels of exogenous androgens inhibit, via a hypothalamic and pituitary negative feedback, gonadotropin and testosterone production. The lack of LH and the consequent functional arrest of Leydig cells determines a marked reduction of intratesticular testosterone, of which adequate concentrations are required to maintain normal spermatogenesis [150]. Furthermore, FSH, in physiological condition, stimulates Sertoli cell to secrete androgen binding protein that conveys testosterone in the lumen of seminiferous tubules. The lack of FSH due to AAS-induced suppression thus aggravates the intratesticular hypotestosteronemia and its consequences on spermatogenesis [151].

Several studies demonstrated that bodybuilders using AAS have a reduced sperm concentration and a higher prevalence of azoospermia compared to controls [152,153]. Current AAS abusers have smaller testicular volume than former AAS abusers and controls, and exhibit markedly decreased plasma gonadotropins, SHBG, 17-hydroxyprogesterone, serum anti-müllerian hormone (AMH) and inhibin B levels. Serum AMH and inhibin B are Sertoli cells biomarkers and low levels are suggestive of impaired spermatogenesis [148].

As for AAS-induced hypogonadotropic hypogonadism, the alteration of spermatogenesis are usually transient. An integrated multivariate time-to-event analysis of data from 30 studies including about 1,500 subjects who assumed AAS for male contraception, revealed that the median time for sperm to recover to 20 million per mL was 3.4 months. The probability of recovery to 20 million per mL was 67% within 6 months, 90% within 12 months, and 100% within 24 months [154]. Since AAS doses used for male contraception are significantly lower than those used for doping purposes, in abusers even longer time could be required to recover spermatogenesis. In any case, the recovery time depends on the individual characteristics of the subject. It has been hypothesized that men who do not obtain normal sperm concentration following AAS withdrawal, could have unknown fertility alterations before starting AAS consumption [155].

An increase in apoptotic processes has also been implicated in the pathogenesis of oligozoospermia following AAS treatment. In rats treated with nandrolone, a statistically significant decrease in testicular weight and in total epididymal sperm count was observed. Contextually, an increase in TUNEL-positive cells and caspase-3 enzyme activity was recorded, indicating the activation of apoptosis [151].

## 10. Conclusions

Surely, substance abuse can contribute to the increased prevalence of hypogonadism observed in Western countries. However, not all abused drugs have a significant negative impact on testicular function.

A low-moderate alcohol intake would seem not to impair reproductive function [22]. Conversely, in heavy drinkers and alcoholic men, both testosterone production and spermatogenesis are altered with multiple mechanisms. Alcohol inhibits some stages of steroidogenesis, increases the conversion of testosterone in estradiol inducing the enzyme aromatase, suppresses β-LH gene expression and protein release from the pituitary gland, and inhibits hypothalamic GnRH secretion through the increase in β-endorphin-like peptides [9,10,11,14,15]. Furthermore, it has been demonstrated that alcohol induces testicular atrophy and histological regressive changes, enhances the production of ROS, and reduces the anti-oxidant testicular defenses [8,10,29].

Regarding tobacco, it has been demonstrated that both cigarette smoke extract and nicotine increase apoptotic markers in sperm [52,53]. Indeed, four meta-analyses have confirmed that smoking is a risk factor for oligozoospermia [44,49,50,51]. Testicular damage is probably due to smoke-induced oxidative stress, but also a neuroendocrine mechanism of nicotine has been hypothesized [31]. Conversely, tobacco does not decrease testosterone levels in men; rather, it would seem to increase androgen concentrations with mechanisms not entirely clear [43].

Caffeine would not seem to reduce testosterone production nor spermatogenesis in men, even if a pro-oxidant effect has been described for very elevated dosage [69].

Despite animal studies showed a decreased testosterone production after exposition to cannabis, a clear relationship between marijuana use and hypogonadism has not been found in men. However, marijuana users have worse sperm parameters, including sperm total count and concentration, compared to controls in most studies [80].

Data about cocaine are few and contrasting. Overall, cocaine abuse seems not to reduce testosterone levels. However, an odds ratio of 2.1 of having low sperm concentration has been found in cocaine abusers [101]. Some Authors found histological regressive changes and apoptotic markers in both germ and Sertoli cells in rats treated with cocaine [91].

Data about the effects of amphetamines on human testicular function are lacking and those on animals are contrasting. Amphetamine, methamphetamine, and MDMA caused variably positive, negative and no effects on testosterone concentrations in rats [109,110,113]. Regarding spermatogenesis, amphetamines induce apoptosis in murine testicular germ cells [114]. Oxidative stress, overproduction of serotonin and GABA, thermic rise, and decreased expression of progesterone and estradiol receptors in testis have been called into question in the pathogenesis of testicular damage [111,116,119].

Opioids intake is a known cause of hypogonadotropic hypogonadism. Indeed, opioids are able to suppress the hypothalamic-pituitary-gonadal axis, with variable grade and intensity depending on the type of compound [120]. The effects on spermatogenesis are less clear. Only one study showed a higher prevalence of oligozoospermia in opium-addicted males [128]. In rats, histological changes, increased apoptosis, and oxidative stress have been described [137].

The inhibitory effects of AAS on testosterone production and spermatogenesis are also widely known. Anabolic-androgenic steroids exert negative feedback on both pituitary gland and hypothalamus, suppressing the release of gonadotropins and testosterone [143]. Testicular volume decreases and sperm production, no longer supported by adequate intratubular testosterone concentration, drops. These effects are usually reversible, even if a complete recovery of the axis can take more than one year. However, incomplete normalization of testosterone levels and sperm production following AAS withdrawal have also been described [148].

The main mechanisms by which substance abuse interferes with testosterone production and spermatogenesis are summarized in Figure 1 and in Table 1.

The effect of the concomitant abuse of more than one of the substances is unpredictable but often additive. The substance withdrawal in most cases leads to the resolution of the hypogonadism. Indeed, during the evaluation of the patient referring for hypogonadism, to investigate the use of illicit drugs and legal substances such as tobacco and alcohol is mandatory. Especially in younger men, when no organic causes of hypogonadism are detectable, substance abuse must be suspected and, eventually, the withdrawal must be recommended.

## Figures and Tables

**Figure 1 jcm-08-00732-f001:**
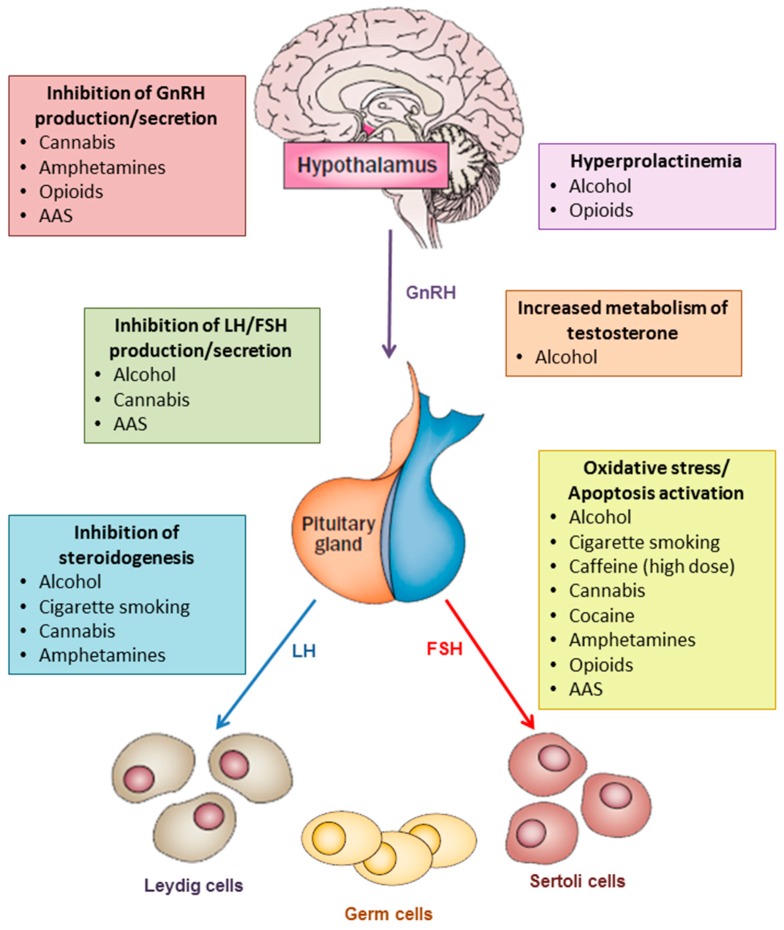
The main mechanisms by which substance/drug abuse may decrease testosterone levels and sperm production are: inhibition oh GnRH production/secretion, increase in prolactin levels, inhibition of gonadotropin production/secretion, inhibition of steroidogenesis, increase in testosterone metabolism, oxidative stress, induction of apoptosis. Abbreviations: AAS = anabolic-androgenic steroids, FSH = follicle-stimulating hormone; GnRH = gonadotropin-releasing hormone; LH = luteinizing hormone.

**Table 1 jcm-08-00732-t001:** Effects of each substance/drug on testosterone levels and sperm production, and main evidences of the pathophysiological mechanisms.↑: increase in testosterone/sperm concentration; ↓: decrease in testosterone/sperm concentration; ↔: no effects on testosterone/sperm concentration; FSH: follicle-stimulating hormone; GABA: gamma-aminobutyric acid; GnRH: gonadotropin-releasing hormone, LH: luteinizing hormone; ROS: reactive oxygen species; StAR: steroidogenic acute regulatory protein

Substance	Effect on Testosterone	Hypothesized Mechanisms	Effect on Sperm Concentration	Hypothesized Mechanisms
**Alcohol**	↓	Suppression of β-LH gene expression [14] Prolactin increase after acute ingestion [9] Inhibition of 3β-hydroxysteroid dehydrogenase and 17-ketosteroid reductase [9] Suppressed expression of StAR via ROS [10] Induction of the enzyme aromatase [11]	↓	Induction of apoptosis [10] Pro-oxidant effect [29]
**Cigarette smoking**	↑	Enhanced GnRH or LH release [38,41] Inhibition of prolactin release [38] Competitive inhibition of testosterone glucuronidation [43]	↓	Induction of apoptosis [52,53] Pro-oxidant effect [56,57,58]
**Caffeine**	↑	Induction of a stress-like hormonal pattern [64]	↔	Pro-oxidant effect at very high doses [69]
**Cannabis**	↔	Inhibition of GnRH and LH in animal models [81,83] Reduced expression of LH receptor on testis in animal models [86] Reduced activity of testicular 3β-hydroxysteroid dehydrogenase in animal models [86]	↓	Induction of apoptosis [71]
**Cocaine**	↔	Panhypopituitarism for pituitary infarction or inflammation (case reports) [96,97]	↓	Testicular vasoconstriction and ischemia [99] Induction of apoptosis [102] Pro-oxidant effect (reperfusion injury) [103]
**Amphetamines**	↓	Decreased expression of GnRH mRNA [112] Activation of adenylate cyclase [106] Inhibition of 3b-hydroxysteroid dehydrogenase, P450c17, and 17-ketosteroid reductase [107] Reduced Ca^2^+ influx [107] Increased testicular GABA concentration [111]	↓	Induction of apoptosis [114,115,116] Pro-oxidant effect [110] Testicular thermic damage [119] Increased testicular serotonin concentration [115] Increased testicular GABA concentration [111] Reduced testicular expression of progesterone and estrogen receptors [116]
**Opioids**	↓	Inhibition of GnRH secretion [120,134] Hyperprolactinemia [120]	↓	Induction of apoptosis [128,140] Pro-oxidant effect [128,129,130,131,132,133,134,135,136,137,138,139,140]
**Anabolic-androgenic steroids (AAS)**	↓	Inhibition of GnRH secretion [143] Inhibition of LH and FSH secretion [145] Depletion of Leydig cells [149]	↓	Reduction of intra-testicular testosterone levels [150,151] Induction of apoptosis [151]

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
