# Peer review of "Substance Abuse and Male Hypogonadism"

_jcm, 2019, doi:10.3390/jcm8050732_

Round 1
Reviewer 1 Report
This review gathers the availabel information on the impact of recreational substance abuse on male gonadal function.
This is a timely and important clinical need as a critical appraisal on the available knowledge on this topic was previously scattered and this will be a useful tool for the practicing clinician.
However, in order to make this document more valuable for the clinical practice this could be improved by the inclusion on tables summarizing the large amount of information provided in the text of this manuscript.
A table should focus on the effect of each drug on the hipothalamic pituitary testicular axis, namely listing the molecular mechanism if already identified and possible alterations found on gonadotropins and testosterone levels in humans.
Another table should focus on the effects of each drug on the spermatogenic function, as this is not necessarily disturbed by the same mechanism which lead to hormonal imbalances. Information on whether the knowledge derived from human or animal models data should included.
The paper needs careful proof reading as despite it is correctly written in English there are a few typographical errors that should be corrected.
Author Response
Attached Rebuttal Letter

Reviewer 2 Report
1. The authors should more clearly stress the diagnosis "Hypogonadism" on laboratory data from a practical point of view.
2. What is the definition of testosteone deficiency (in mg/dl and in nmol/l)?
3. The SHBG as the major Testost. binding protein is only briefly mentioned on page 1 line 44 and on page 3 line 107.and later on page 12. So, SHBG has a major impact on Testost. concentrations, This should be mentioned and also the problems with free Testosterone measurements.
1-3. should be included in the Introduction.
-Line 60: This review is not "briefly".
-Testosterone is converted to Estradiol in patients with liver disease. The clinical consequence (Gynekomastia) should be mentioned.
-Testostetone and AAS also increases Hkt! l
- Hypogonadism is a more prevalent clinical problem than spermatogenesis. The authors should .mention the impact of hypogonadism (osteoporosis, mood disorders, muscular problems.
Author Response
Attached Rebuttal Letter